# The Hospital Anxiety and Depression Scale (HADS) applied to Ethiopian cancer patients

**Yemataw Wondie[1], Anja Mehnert[2], Andreas Hinz[2]***

**1** Department of Psychology, University of Gondar, Gondar, Ethiopia, **2** Department of Medical Psychology and Medical Sociology, University of Leipzig, Leipzig, Germany

* Andreas.Hinz@medizin.uni-leipzig.de

## Abstract

Psychological distress is a common problem associated with cancer. The main objective of the present study was to test the Hospital Anxiety and Depression Scale (HADS) in a sample of Ethiopian cancer patients and to compare the results with those obtained from a sample in Germany. Data were collected from 256 cancer patients who visited the University of Gondar Hospital between January 2019 and June 2019 using the HADS, the European Organization for Research and Treatment of Cancer Quality of Life questionnaire (EORTC QLQ-C30), and the Multidimensional Fatigue Inventory MFI-20. The reliability of the HADS was good, with Cronbach's α coefficients of 0.86 (anxiety), 0.85 (depression), and 0.91 (total scale). The Ethiopian cancer patients were more anxious (M = 7.9) and more depressed (M = 9.3) than the German patients (M = 6.8 for anxiety and M = 5.5 for depression). Only a weak level of measurement invariance was detected between the Ethiopian and the German sample. In the Ethiopian sample, anxiety and depression were associated with tumor stage (high levels in stage 4) and treatment (high levels for patients not receiving surgery and chemotherapy). Both anxiety and depression were significantly associated with all of the EORTC QLQ-C30 and MFI-20 scales. The HADS proved to be applicable for use with Ethiopian cancer patients. The high level of anxiety and depression present in that group indicates a need for psychosocial care.

## Introduction

Cancer incidence and mortality rates are lower in low-income countries (LICs) than in high-income countries (HICs). Of the 7.6 billion people alive on earth in 2018, about 1.3 billion people (17%) resided in Africa. As far as cancer incidence rates are concerned, Africa accounts for only 4.7 percent of cases worldwide, and the incidence of cancer mortality (5.9%) there is only slightly higher [1]. Nevertheless, cancer incidence and mortality rates are increasing in LICs such as Ethiopia, mainly due to increases in life expectancy [2–7]. Mental health care for oncological patients is however not well established in LICs, and knowledge about the status of patients' mental health in those countries is very limited. Nonetheless, some studies have been performed to investigate mental health and quality of life (QoL) in specific samples of Ethiopian cancer patients. Most of these were conducted in the capital, Addis Ababa, among patients suffering from breast or gynecologic cancer [8–11].

**Competing interests:** The authors have declared that no competing interests exist.

Anxiety and depression are common psychological symptoms in cancer patients [12–15]. These symptoms can affect QoL, adherence to treatment, cancer survival, and treatment costs [16–18]. The Hospital Anxiety and Depression Scale (HADS), developed by Zigmond and Snaith [19], is one of the most frequently used instruments for measuring anxiety and depression in patients suffering from physical illness, including cancer. It has been applied to a wide range of diseases [20], and multiple studies have been performed to test psychometric properties of the scale [21–24]. Most of the psycho-oncological studies that have used the HADS have been conducted in HICs. A particular problem in LICs is illiteracy. In Ethiopia, where the literacy rate is low, many patients cannot fill in questionnaires without help. In previous Ethiopian studies done on QoL in cancer patients [8,10,25], more than 50% of the patients were illiterate or never went to school. In such cases, research assistants must read the questions aloud, ask the patients to respond verbally, and mark the response in the questionnaire. It has not yet been systematically studied whether this method of data collection has a substantial impact on the outcome.

Measurement invariance of the instrument used is an important issue to consider when comparing the mean scores of different samples. To our knowledge, measurement invariance of the HADS had never previously been tested in a comparison between a HIC and a LIC. We therefore also tested the measurement invariance of the HADS by comparing an Ethiopian and a German sample of cancer patients.

Multiple examinations have been performed to study the impact of sociodemographic and clinical variables on anxiety and depression [15,26–28]. One typical result was that female cancer patients were generally more anxious than male patients [15,27,29]. Most of these studies have also been performed in HICs, and it is unclear to what degree the results are also applicable to LICs.

The general objective of this study was to test psychometric properties of the HADS in a sample of Ethiopian cancer patients and to compare the results with those obtained in Germany, a country with a longer tradition of applying this questionnaire. In particular, the aims were (a) to explore the degree of anxiety and depression in Ethiopian cancer patients in comparison with German cancer patients, (b) to analyze the impact of sociodemographic and clinical variables on anxiety and depression, (c) to test psychometric properties of the HADS, (d) to compare the applicability of the scale in the Ethiopian context with the German context via measurement invariance analysis, and (e) to analyze the relationship between the HADS scales and several facets of QoL.

## Methods

### Ethiopian cancer patients

The present study was conducted at the University of Gondar Specialized Hospital, Ethiopia. Gondar is the second largest city in Ethiopia, with a population of about 300,000 inhabitants. Between January 2019 and June 2019, 298 cancer patients treated at this hospital were eligible for this study. Participants were invited to take part in the study if they had a malignant tumor of any cancer site and disease stage, were at least 18 years old, and understood Amharic, the national language of Ethiopia and that predominantly spoken in Gondar and its surroundings. Tumor entities, disease stage, and illiteracy did not serve as exclusion criteria. Trained nurse research assistants contacted the patients, explained the aims of the study, and asked them to participate and give informed consent. If the patients were illiterate, the research assistants read the questions aloud, asked the patients to respond verbally, and marked the response in the questionnaire. Medical data was taken from the medical records kept by the hospital. The study was conducted in accordance with the Declaration of Helsinki and was approved by the

Institutional Review Board of the University of Gondar (Ref. No. O/V/P/RCS/05/1542/2018; dated June 18, 2018).

## German cancer patients

A group of German cancer patients served as controls. The patients were recruited in five study centers in Germany. Further details of the sample are published elsewhere [30]. Data on anxiety and depression in this sample have already been published [28]. The sample consisted of 1,821 males and 1,964 females with a mean age of 58.3 years. Because of the differences in the age and gender distribution between the Ethiopian and the German samples, we selected a subsample of the German cancer patients matched to the Ethiopian one in those aspects. The resulting German subsample consisted of 1,664 cancer patients, of which 638 (38.3%) and 1,026 (61.7%) were males and females, respectively. The mean age of this group was 48.0 years. All participants gave informed consent. This study was also conducted in accordance with the Declaration of Helsinki and was approved by the Ethics Committees of each of the five participating universities: the Ethics Committee of the University of Leipzig, the Ethics Committee of the University of Hamburg, the Ethics Committee of the University of Freiburg, the Ethics Committee of the University of Heidelberg, and the Ethics Committee of the University of Würzburg.

## Questionnaires

The HADS consists of 14 items. Seven of the items indicate anxiety and the remaining seven items indicate depression. The answer format offers four response options, which are scored with values ranging from 0 to 3. This results in scale values between 0 and 21 for each scale. The original test authors defined three ranges for both of the scales: 0–7 (non-cases), 8–10 (doubtful cases), and 11–21(cases). It is possible to calculate a HADS total score by simply summing up the anxiety and depression items [31]. Normative values are available for several countries [32–35]. The Amharic translation of the HADS was adopted from a study conducted among Ethiopian HIV patients [36].

Two other questionnaires were used in addition to the HADS. The European Organization for Research and Treatment of Cancer Quality of Life questionnaire (EORTC QLQ-C30) [37] is comprised of 30 items which are assigned to five functioning scales (physical, role, emotional, social, and cognitive functioning), three symptom scales (fatigue, pain, and nausea/vomiting), a two-item global health /QoL scale, and six single item scales (dyspnea, appetite loss, insomnia, constipation, diarrhea, and financial difficulties). Higher functioning scores represent better functioning/QoL, whereas higher symptom scores represent more severe symptoms. A sum score of the EORTC QLQ-C30 can be calculated, following a recommendation of the EORTC Quality of Life Group [38]. The Multidimensional Fatigue Inventory (MFI-20) [39] was used to measure fatigue. It assesses five dimensions of fatigue: general fatigue, physical fatigue, reduced activity, reduced motivation, and mental fatigue. Each subscale consists of four items, scored with values from 1 to 5. Higher scores reflect a higher level of fatigue. We also calculated a sum score for all 20 items.

## Statistical analysis

Mean score differences were expressed in terms of Cohen's effect sizes *d*. Reliability was measured with Cronbach's coefficient α. To test the impact of sociodemographic and clinical variables on anxiety and depression, we used three-factor ANOVAs with gender and age group as cofactors. Confirmatory Factor Analyses (CFAs) were calculated to test the two-dimensional structure of the HADS. The measurement invariance between the Ethiopian and the German

data set was analyzed with measurement invariance analyses using the criteria $Chi^2/df$, Comparative Fit Index (CFI), Tucker-Lewis Index (TLI), Root Mean Square Error of Approximation (RMSEA), and Standardized Root Mean Square Residual (SRMR). The associations between the HADS scales and scales of other questionnaires were expressed with Pearson correlations. The CFA statistics were performed with Mplus, and all other statistics were calculated with SPSS version 24.

## Results

### Sociodemographic characteristics of the Ethiopian sample

Of the 298 patients eligible for participation in the study, 256 (85.9%) were gave informed consent and filled in the questionnaires. The research assistants proofed the questionnaires so that there remained no missing items in the questionnaires. Of the 256 participants, 135 (52.7%) were illiterate. Further sociodemographic and clinical variables are given in Table 1. The most frequent cancer types were breast cancer (25.4%), colon cancer (17.2%), and non-Hodgkin lymphoma (14.5%).

In the German sample, the most frequent cancer localizations were: breast (29.0%), digestive organs (18.8%), female genital organs (9.9%), blood and blood forming organs (9.1%), male genital organs (8.1%), and respiratory organs (7.4%). The distribution of the tumor stage in the German sample was: stage 1 (22.2%), stage 2 (24.3%), stage 3 (20.9%), and stage 4 (32.6%).

### Psychometric analyses on scale level and item level

The right part of Table 2 shows (part-whole-corrected) item-test correlations for the items and reliability coefficients (Cronbach's $\alpha$) for the scales. All $\alpha$ coefficients of the Ethiopian sample were good ($\alpha \geq 0.85$) end even slightly higher than the coefficients obtained from the German sample. All items contributed positively to the scale scores with coefficients between 0.53 and 0.67.

The comparison between the Ethiopian and the German mean scores indicates that the Ethiopian patients had higher levels of psychological burden than the German patients. This difference was especially high in the depression scale. All but one of the 14 items showed higher means in the Ethiopian sample. One item (A3: frightened feeling) showed an opposite trend. The greatest difference between the Ethiopian and the German cancer patients was found for the depression item D14, whereby an effect size of greater than 1 was observed. The distributions of the three categories (no, doubtful, severe cases of anxiety and depression) in the two samples are presented in Table 3.

### Measurement invariance between the Ethiopian and German samples' HADS scores

The results of two-factor CFA of the Ethiopian sample are illustrated in Fig 1. The latent variables of anxiety and depression were correlated with $r = 0.88$, whereas the correlation of the mean scores of anxiety and depression was $r = 0.75$. All item loadings were between 0.55 and 0.76, the lowest loading was found for item D14 (book/TV), with a coefficient of 0.55.

The fit indices are given in the upper part of Table 4. CFI and TLI were higher than 0.90 for both the Ethiopian and the German sample.

While the criteria for metric (weak) invariance were fulfilled, scalar (strong) invariance was not supported since the CFI and TLI coefficients were lower than 0.90, and SRMR was higher than 0.10. The model fit for strict invariance was even less satisfying than that for the strong invariance. The lower part of Table 4 documents the differences in the fit indices between subsequent models.

**Table 1. Characteristics of the sample of Ethiopian cancer patients.**

| | Males (n = 99) | | Females (n = 157) | | Total (n = 256) | |
|---|---|---|---|---|---|---|
| | *n* | % | *n* | % | *n* | % |
| Age (years) | | | | | | |
| M (SD) | 51.6 | 15.8 | 45.6 | 13.3 | M = 47.9 | SD = 14.6 |
| Age category | | | | | | |
| 18–49 y. | 37 | 37.4 | 90 | 57.3 | 127 | 49.6 |
| ≥ 50 y. | 62 | 62.6 | 67 | 42.7 | 129 | 50.4 |
| Marital status | | | | | | |
| Single | 16 | 16.2 | 23 | 14.6 | 39 | 15.2 |
| Married | 76 | 76.8 | 82 | 52.2 | 158 | 61.7 |
| Divorced | 7 | 7.1 | 30 | 19.1 | 37 | 14.5 |
| Separate /Widowed | 0 | 0.0 | 22 | 14.0 | 22 | 8.6 |
| Education | | | | | | |
| Illiterate | 48 | 48.5 | 87 | 55.4 | 135 | 52.7 |
| Elementary school | 19 | 19.2 | 19 | 12.1 | 38 | 14.8 |
| Secondary school | 13 | 13.1 | 18 | 11.5 | 31 | 12.1 |
| High school | 5 | 5.1 | 7 | 4.5 | 12 | 4.7 |
| Technical and vocational college | 9 | 9.1 | 16 | 10.2 | 25 | 9.8 |
| University | 5 | 5.1 | 10 | 6.4 | 15 | 5.9 |
| Religion | | | | | | |
| Christian | 89 | 89.9 | 141 | 89.8 | 230 | 89.8 |
| Muslim | 10 | 10.1 | 16 | 10.2 | 26 | 10.2 |
| Tumor | | | | | | |
| Breast | 2 | 2.0 | 63 | 40.1 | 65 | 25.4 |
| Colon | 24 | 24.2 | 20 | 12.7 | 44 | 17.2 |
| Non-Hodgkin lymphoma | 22 | 22.2 | 15 | 9.6 | 37 | 14.5 |
| Cervix uteri | 0 | 0.0 | 15 | 9.6 | 15 | 5.9 |
| Corpus uteri | 0 | 0.0 | 9 | 5.7 | 9 | 3.5 |
| Prostate | 9 | 9.1 | 0 | 0.0 | 9 | 3.5 |
| Colorectal | 3 | 3.0 | 5 | 3.2 | 8 | 3.1 |
| Thyroid | 2 | 2.0 | 6 | 3.8 | 8 | 3.1 |
| Lymphocytic lymphoma | 2 | 2.0 | 4 | 2.5 | 6 | 2.3 |
| Pancreas | 5 | 5.1 | 1 | 0.6 | 6 | 2.3 |
| Lung | 4 | 4.0 | 2 | 1.3 | 6 | 2.3 |
| Other | 26 | 26.3 | 17 | 10.8 | 43 | 16.8 |
| Tumor stage, UICC [a] | | | | | | |
| 1 | 5 | 5.1 | 15 | 9.6 | 20 | 7.8 |
| 2 | 20 | 20.2 | 46 | 29.3 | 66 | 25.8 |
| 3 | 22 | 22.2 | 41 | 26.1 | 63 | 24.6 |
| 4 | 26 | 36.4 | 44 | 28.0 | 80 | 31.3 |
| Surgery | | | | | | |
| No | 56 | 56.6 | 69 | 43.9 | 125 | 48.8 |
| Yes | 43 | 43.4 | 88 | 56.1 | 131 | 51.2 |
| Radiation | | | | | | |
| No | 90 | 90.9 | 144 | 91.7 | 234 | 91.4 |
| Yes | 9 | 9.1 | 13 | 8.3 | 22 | 8.6 |
| Chemotherapy | | | | | | |

(*Continued*)

**Table 1.** (Continued)

| | Males | | Females | | Total | |
|---|---|---|---|---|---|---|
| | (*n* = 99) | | (*n* = 157) | | (*n* = 256) | |
| | *n* | % | *n* | % | *n* | % |
| No | 46 | 46.5 | 63 | 40.1 | 109 | 42.6 |
| Yes | 53 | 53.5 | 94 | 59.9 | 147 | 57.4 |

Note.
[a] Missing data not reported.

## The impact of sociodemographic and clinical variables on anxiety and depression

Table 5 shows the impact of sociodemographic and clinical variables on anxiety and depression in the Ethiopian sample. Neither age nor gender was significantly associated with anxiety and depression. Patients with low levels of education were more depressed than their better-educated counterparts. There was a nearly linear increase of anxiety and depression with tumor stage. Patients who received surgery or chemotherapy were less anxious and less depressed than patients who were not receiving cancer treatment (Table 5).

## Comparison between illiterate and literate patients

Since education had a significant impact on depression, we looked for the items that contributed most significantly to this difference. To do so, we calculated the 14 item mean scores for the 135 illiterate patients and those 121 patients with school education separately, and we

**Table 2. HADS mean scores and psychometric criteria, comparison between Ethiopia and Germany.**

| | Ethiopia | | Germany | | Effect | Ethiopia | Germany |
|---|---|---|---|---|---|---|---|
| Items | M | (SD) | M | (SD) | size *d* | $r_{it}$ | $r_{it}$ |
| A1: Tense, wound up | 1.23 | (0.88) | 1.07 | (0.76) | 0.20 | 0.65 | 0.57 |
| A3: Frightened feeling | 1.19 | (0.96) | 1.27 | (0.96) | -0.08 | 0.64 | 0.61 |
| A5: Worries | 1.14 | (0.97) | 1.04 | (0.83) | 0.11 | 0.64 | 0.66 |
| A7: Relaxed | 1.20 | (0.93) | 0.99 | (0.81) | 0.24 | 0.62 | 0.61 |
| A9: Butterflies | 1.04 | (0.80) | 0.90 | (0.72) | 0.18 | 0.65 | 0.60 |
| A11: Restless | 1.08 | (0.88) | 0.98 | (0.90) | 0.11 | 0.64 | 0.33 |
| A13: Panic | 0.98 | (0.80) | 0.57 | (0.72) | 0.54 | 0.60 | 0.63 |
| D2: Enjoy things | 1.39 | (1.00) | 0.77 | (0.79) | 0.69 | 0.67 | 0.70 |
| D4: Laugh/funny side | 1.33 | (0.92) | 0.66 | (0.75) | 0.80 | 0.61 | 0.72 |
| D6: Cheerful | 1.04 | (0.83) | 0.78 | (0.85) | 0.31 | 0.65 | 0.69 |
| D8: Slowed down | 1.69 | (1.01) | 1.41 | (0.83) | 0.30 | 0.61 | 0.49 |
| D10: Appearance | 1.11 | (0.97) | 0.43 | (0.73) | 0.80 | 0.58 | 0.43 |
| D12: Enjoyment | 1.11 | (0.98) | 0.99 | (0.91) | 0.13 | 0.67 | 0.67 |
| D14: Book/TV | 1.68 | (1.13) | 0.48 | (0.70) | 1.31 | 0.53 | 0.56 |
| Scales | | | | | | | |
| Anxiety | 7.9 | (4.6) | 6.8 | (4.0) | 0.26 | α = 0.86 | α = 0.82 |
| Depression | 9.3 | (5.0) | 5.5 | (4.1) | 0.84 | α = 0.85 | α = 0.85 |
| Total score | 17.2 | (9.0) | 12.3 | (7.4) | 0.60 | α = 0.91 | α = 0.89 |

Note. *d*: Effect size of the difference between the Ethiopian and the German mean scores; $r_{it}$: Part-whole-corrected item-test correlations.

**Table 3. Categorical distribution of anxiety and depression.**

| | Ethiopia | | | | Germany | | | |
|---|---|---|---|---|---|---|---|---|
| | Anxiety | | Depression | | Anxiety | | Depression | |
| | n | % | n | % | N | % | n | % |
| No cases | 129 | 50.4 | 85 | 33.2 | 995 | 59.8 | 1172 | 70.4 |
| Doubtful cases | 48 | 18.7 | 61 | 23.8 | 337 | 20.3 | 268 | 16.1 |
| Severe cases | 79 | 30.9 | 110 | 43.0 | 332 | 20.0 | 224 | 13.5 |

expressed the group differences in terms of effect sizes (Table 6). All of the illiterate subgroup's item mean scores were equal or higher than those of the other group. The effect sizes of the seven anxiety items ranged from 0.00 to 0.32 and from 0.13 to 0.79 for the depression items. The highest differences were found for item D14 (books/TV) ($d = 0.79$), while all of the other items had effect sizes below 0.50. Item D14 was also the item with the lowest loadings in Fig 1 and the largest group differences between the Ethiopian and the German cancer patients (Table 2). It is understandable that enjoyment of reading books and watching TV depends on a person's ability to read and their access to TV devices. Therefore, we also calculated several analyses excluding this item. When the depression scale was based on the remaining six items, Cronbach's α coefficient remained unchanged ($α = 0.85$), and the effect size for the depression difference between Ethiopia and Germany reduced from $d = 0.84$ to $d = 0.66$. When we allowed the intercept of item D14 to be different in the measurement invariance analysis (Table 4), scalar (strong) invariance was approached. Based on the assumption that the other 13 items have equal intercepts and only item D14 was allowed to have different parameters, the following coefficients were obtained: $Chi^2(175) = 1302.92$, $Chi^2/df = 7.445$, $CFI = 0.901$, $TLI = 0.897$, and $RMSEA = 0.058$.

Illiteracy was confounded with several variables. Among the illiterate patients the proportion of older patients was higher ($p = 0.001$), and the proportions of patients receiving surgery and chemotherapy was lower ($p = 0.002$ and $p = 0.004$, respectively) than in the group of

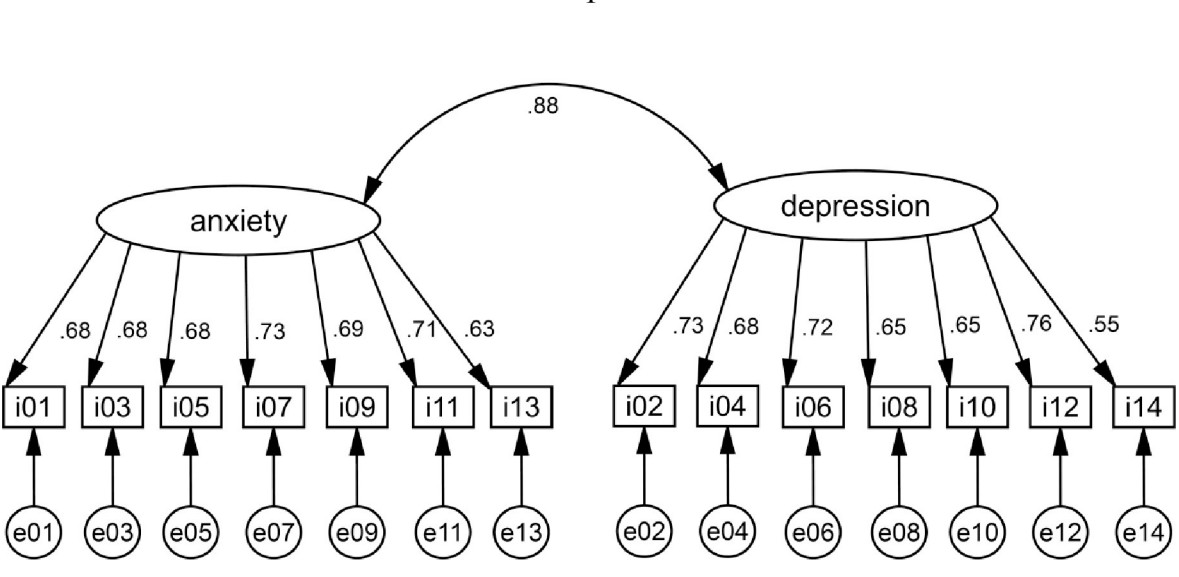

**Fig 1. CFA results for the Ethiopian sample.**

**Table 4. Results of the measurement invariance analyses.**

| Level of invariance | Ethiopia vs. Germany | Chi² (df) | Chi²/df | CFI | TLI | RMSEA | SRMR |
|---|---|---|---|---|---|---|---|
| Configural invariance | Ethiopia (N = 256) | 204.80 (76) | 2.70 | 0.919 | 0.903 | 0.082 | 0.052 |
| | Germany (N = 1664) | 756.40 (76) | 9.95 | 0.930 | 0.917 | 0.073 | 0.043 |
| Baseline | Unconstrained | 961.46 (152) | 6.33 | 0.929 | 0.915 | 0.053 | 0.052 |
| Metric (weak) invar. | Weights fixed | 1053.59 (166) | 6.35 | 0.922 | 0.914 | 0.053 | 0.117 |
| Scalar (strong) invar. | Weights and intercepts fixed | 1452.0 (176) | 8.25 | 0.888 | 0.884 | 0.061 | 0.101 |
| Full (strict) invar. | All parameters fixed | 2059.8 (191) | 10.78 | 0.835 | 0.843 | 0.071 | 0.111 |
| **Fit differences** | | | | | | | |
| Level of invariance | Model comparison | | | ΔCFI | ΔTLI | ΔRMSEA | ΔSRMR |
| Metric (weak) invar. | Weights fixed vs. baseline | | | -0.007 | -0.001 | 0.000 | 0.065 |
| Scalar (strong) invar. | Weights and intercepts fixed vs. weights fixed | | | -0.034 | -0.030 | 0.008 | -0.016 |
| Full (strict) invar. | All parameters fixed vs. weights and intercepts fixed | | | -0.053 | -0.041 | 0.010 | 0.010 |

literate patients. There were no statistically significant associations between illiteracy and gender, marital status, tumor type, and tumor stage.

## Correlations with other scales

Correlations between the HADS scores and the scales of the EORTC QLQ-C30 and the MFI-20 are presented in Table 7. Despite the peculiarity of item D14, we used the original scales with seven items each. Anxiety and depression were most strongly correlated with the MFI-20 fatigue sum score and with the sum score of the EORTC QLQ-C30. The correlations were very similar for the anxiety and depression subscale of the HADS.

## Discussion

The first research question was whether the Amharic version of the HADS showed acceptable psychometric properties. The reliability was good with Cronbach's α coefficients of 0.86/0.85/0.91 for the HADS anxiety subscale, depression subscale, and the total scale, respectively. These coefficients were even slightly higher than those of our German comparison sample (0.82/0.85/0.89) and in other studies performed in Mexico (0.79/0.80/0.86) [40], Chile (0.76/0.84/0.87) [41], Greece (0.83/0.84/0.88) [42], Australia (0.90/0.86/ 0.92) [43], and among Latina women in the US (0.85/0.83/0.88) [24].

All corrected item-test correlations were above 0.50 in the Ethiopian cancer patients' sample, which means that each item positively contributed to the scale scores. Taken together, these reliability coefficients suggest the applicability of the Amharic version of the HADS for use in groups of cancer patients. However, the CFA coefficients of the Ethiopian sample were somewhat weaker than those of the German sample. Since the Ethiopian Cronbach α coefficients were nevertheless high, this means that the Ethiopian patients did not differentiate between the subscales anxiety and depression as sharply as the German patients did, an interpretation which is also supported by the high correlation ($r = 0.75$) between the anxiety and depression subscales.

***Measurement invariance between the Ethiopian and the German data set was established on a weak level (equal loadings) but not on the level of strong invariance (equal intercepts). One reason for this kind of differential item functioning was that the item D14 ("I can enjoy a good book or radio or TV program") performed in different ways in Ethiopia and Germany. The loading of this item was sufficiently high (0.55) even in the Ethiopian sample, but the intercepts were different which means that fewer Ethiopian patients agreed that they

**Table 5. HADS mean scores of the Ethiopian sample, depending on sociodemographic and clinical variables.**

| | n | Anxiety | | Depression | |
|---|---|---|---|---|---|
| | | M | (SD) | M | (SD) |
| Gender | | | | | |
| Male | 99 | 8.0 | (4.8) | 9.5 | (4.7) |
| Female | 157 | 7.8 | (4.5) | 9.2 | (5.2) |
| (Significance) | | (F = 0.328) | (p = 0.568) | (F = 0.050) | (p = 0.824) |
| Age group | | | | | |
| ≤ 49 years | 127 | 8.0 | (4.7) | 8.9 | (5.2) |
| ≥ 50 years | 129 | 7.7 | (4.6) | 9.8 | (4.7) |
| (Significance) | | (F = 0.829) | (p = 0.363) | (F = 1.836) | (p = 0.177) |
| Marital status | | | | | |
| Single | 39 | 8.5 | (4.2) | 9.4 | (4.8) |
| Married | 158 | 7.7 | (4.8) | 9.2 | (5.0) |
| Divorced, widowed | 59 | 8.0 | (4.4) | 9.7 | (5.0) |
| (Significance) | | (F = 0.325) | (p = 0.723) | (F = 0.446) | (p = 0.640) |
| Education | | | | | |
| Illiterate | 135 | 8.4 | (4.5) | 10.5 | (4.8) |
| Elem. and sec. school | 69 | 7.0 | (4.7) | 8.5 | (4.8) |
| High school and above | 22 | 7.9 | (4.7) | 7.1 | (5.0) |
| (Significance) | | (F = 1.907) | (p = 0.151) | (F = 8.523) | (p<0.001) |
| Tumor type | | | | | |
| Breast | 65 | 6.3 | (4.6) | 7.9 | (5.6) |
| Colon | 44 | 7.2 | (4.1) | 8.7 | (4.4) |
| Non-Hodgkin lymphoma | 37 | 9.2 | (5.0) | 10.3 | (5.2) |
| Others | 110 | 8.6 | (4.4) | 10.1 | (4.5) |
| (Significance) | | (F = 2.322) | (p = 0.076) | (F = 1.215) | (p = 0.305) |
| Stage | | | | | |
| 1 | 20 | 7.0 | (3.5) | 8.4 | (3.7) |
| 2 | 66 | 6.7 | (4.2) | 8.5 | (4.4) |
| 3 | 63 | 7.5 | (5.0) | 8.2 | (5.3) |
| 4 | 80 | 8.5 | (4.7) | 10.5 | (5.0) |
| (Significance) | | (F = 2.084) | (p = 0.103) | (F = 3.313) | (p = 0.021) |
| Surgery | | | | | |
| No | 125 | 9.2 | (4.5) | 10.9 | (4.7) |
| Yes | 131 | 6.6 | (4.4) | 7.8 | (4.8) |
| (Significance) | | (F = 22.869) | (p<0.001) | (F = 20.587) | (p<0.001) |
| Chemotherapy | | | | | |
| No | 109 | 8.7 | (4.4) | 10.3 | (4.9) |
| Yes | 147 | 7.3 | (4.7) | 8.6 | (4.9) |
| (Significance) | | (F = 6.269) | (p = 0.013) | (F = 6.732) | (p = 0.010) |

enjoyed reading books and watching TV. This illustrates that the transfer of questionnaires from Western countries to low income countries may be problematic [44,45]. However, removing item D14 and calculating analyses with the remaining six items has the disadvantage that the results cannot be compared with those reported from other studies. Hence, we preferred to keep all of the scales.

The comparison between the Ethiopian and the German cancer patients showed that the Ethiopian patients were more anxious (d = 0.26) and more severely depressed (d = 0.84) than

**Table 6. Item mean scores for illiterate and literate patients.**

| | Illiterate | | Literate | | Effect size *d* |
|---|---|---|---|---|---|
| | **M** | **(SD)** | **M** | **(SD)** | |
| Items | | | | | |
| A1: Tense, wound up | 1.33 | (0.92) | 1.12 | (0.82) | 0.24 |
| A3: Frightened feeling | 1.23 | (0.92) | 1.15 | (1.00) | 0.08 |
| A5: Worries | 1.27 | (0.96) | 0.98 | (0.97) | 0.30 |
| A7: Relaxed | 1.33 | (0.95) | 1.04 | (0.89) | 0.32 |
| A9: Butterflies | 1.04 | (0.72) | 1.04 | (0.88) | 0.00 |
| A11: Restless | 1.13 | (0.87) | 1.03 | (0.90) | 0.11 |
| A13: Panic | 1.02 | (0.76) | 0.94 | (0.84) | 0.10 |
| D2: Enjoy things | 1.55 | (0.97) | 1.21 | (1.02) | 0.34 |
| D4: Laugh/funny side | 1.42 | (0.92) | 1.22 | (0.92) | 0.22 |
| D6: Cheerful | 1.09 | (0.76) | 0.98 | (0.89) | 0.13 |
| D8: Slowed down | 1.91 | (1.00) | 1.44 | (0.97) | 0.48 |
| D10: Appearance | 1.22 | (0.99) | 0.98 | (0.93) | 0.25 |
| D12: Enjoyment | 1.27 | (0.91) | 0.94 | (1.03) | 0.34 |
| D14: Book/TV | 2.07 | (1.01) | 1.24 | (1.09) | 0.79 |
| Scales | | | | | |
| Anxiety | 8.36 | (4.51) | 7.31 | (4.69) | 0.23 |
| Depression | 10.53 | (4.79) | 8.02 | (4.87) | 0.52 |
| Total score | 18.90 | (8.69) | 15.33 | (8.95) | 0.40 |

the German ones. The HADS mean scores of the Ethiopian cancer patients (7.9/9.3) for anxiety and depression, respectively, were also markedly higher than those obtained in other countries such as India (6.5/6.8) [46], Chile (3.9/6.2) [41], and Jordan (M = 6.3/7.9) [47], and the scores are also much higher than those obtained with the Amharic HADS version in Ethiopian HIV patients (4.0/4.0) [36]. Since normative HADS data from Ethiopia are not available, the question must remain open concerning the degree to which the differences between the Ethiopian cancer patients and cancer patients from other countries are due to the particular situation of cancer patients, and to what degree these mean score differences reflect general response tendencies in Ethiopia. Nevertheless, the high levels of mental psychological burden in the Ethiopian patients lead to the conclusion that there are unrecognized and unmet needs for psychosocial care.

Anxiety and depression were associated with tumor stage. Patients with stage 4 cancer had the highest depression scores. This was also observed in other studies performed in Ethiopia [11], Germany [28], Taiwan [48], Mexico [49], and several Southeast Asian countries [50], and can be considered an argument for the known-groups validity of the Amharic version of the HADS.

While the relationship between tumor stage and depression can easily be understood, it is interesting to see that patients who received medical treatment (surgery or chemotherapy) were significantly less anxious and less depressed than patients who did not get such treatment. This is also in line with other studies from low- and middle-income Southeast Asian countries, where patients who received surgery were less anxious and less depressed than those who did not. This relationship is less pronounced in studies performed in HICs. One possible reason is that in LICs, treatment is primarily focused on patients with a lower tumor stage, while patients with stage 4 cancer often do not get such treatment. Another possible reason might be that the fact of getting medical help and treatment itself (irrespective of the medical impact of

**Table 7. Correlations between the HADS scores and scales of other questionnaires.**

| | Anxiety | | Depression | | Total score | |
|---|---|---|---|---|---|---|
| EORTC QLQ-C30 | | | | | | |
| Physical functioning | -.49 | *** | -.57 | *** | -.57 | *** |
| Role functioning | -.53 | *** | -.57 | *** | -.59 | *** |
| Emotional functioning | -.65 | *** | -.57 | *** | -.65 | *** |
| Cognitive functioning | -.60 | *** | -.61 | *** | -.64 | *** |
| Social functioning | -.55 | *** | -.55 | *** | -.59 | *** |
| Global health/QoL | -.50 | *** | -.53 | *** | -.55 | *** |
| Fatigue | .57 | *** | .64 | *** | .65 | *** |
| Nausea/Vomiting | .38 | *** | .35 | *** | .39 | *** |
| Pain | .58 | *** | .62 | *** | .65 | *** |
| Dyspnoea | .40 | *** | .36 | *** | .40 | *** |
| Insomnia | .47 | *** | .43 | *** | .49 | *** |
| Appetite loss | .44 | *** | .44 | *** | .47 | *** |
| Constipation | .31 | *** | .32 | *** | .34 | *** |
| Diarrhea | .17 | ** | .10 | ns | .15 | * |
| Financial difficulties | .39 | *** | .41 | *** | .43 | *** |
| Sum score | -.68 | *** | -.69 | *** | -.73 | *** |
| MFI-20 | | | | | | |
| General fatigue | .58 | *** | .69 | *** | .68 | *** |
| Physical fatigue | .61 | *** | .64 | *** | .67 | *** |
| Reduced activity | .54 | *** | .63 | *** | .63 | *** |
| Reduced motivation | .49 | *** | .61 | *** | .59 | *** |
| Mental fatigue | .61 | *** | .65 | *** | .68 | *** |
| Fatigue sum score | .65 | *** | .74 | *** | .75 | *** |

*: $p < 0.05$

**: $p < 0.01$

***: $p < 0.001$; ns: Not significant.

such treatment) has a greater beneficial effect on the mental situation of patients in LICs countries than it does in HICs.

One finding was that there were differences between illiterate and literate people in their responses to the HADS. The percentage of illiterate patients (52.7%) in this study was comparable to that reported in other Ethiopian examinations: 43.9% [51], 68.6% [25], 69.0% [10], and 80.0% [8]. Although literacy rates are increasing, illiteracy remains a major problem in psycho-oncological research in Ethiopia. Nevertheless, our study showed that the research assistants were still able to get reliable data from illiterate patients. Although it took them a long time to interview the patients and fill in the questionnaire when the study first began, after gaining some experience with the procedure, the study assistants became familiar with the materials and could perform the study more effectively.

In the discussion of the measurement invariance between Ethiopia and Germany, we mentioned that item D14 (reading books/watching TV) strongly contributed to the differences. This item also showed the most pronounced difference between the illiterate and the literate patients within the Ethiopian sample ($d = 0.79$). However, in all of the other items the illiterate patients were more anxious and depressed than the literate ones. Since there were no significant differences between these groups in terms of cancer type and tumor stage, these differences cannot be due to these possible confounders. Further research is needed to investigate

the impact of illiteracy on the responses to questionnaires and to interpret the items in the context of cultural differences [44,45,52].

Anxiety and depression were strongly correlated with all facets of QoL (except diarrhea) and fatigue (Table 7). Though the correlations with the Emotional functioning scale of the EORTC QLQ-C30 were among the highest coefficients, the associations with fatigue and with pain were of nearly equal magnitude. The high correlations underline the importance of anxiety and depression for the assessment of QoL.

Some *limitations* of this study should be mentioned. The Ethiopian patient sample was recruited in one hospital in Gondar; therefore, the generalizability to other Ethiopian or African areas is unclear. The sample size was sufficient for comparisons on the group level, but not large enough for comparisons among various cancer types with sufficient precision. The Ethiopian and German samples were matched for age and gender, but not for clinical variables. Due to the diversity of tumor localizations and the lack of tumor stage data in both samples, it was not possible to match after these clinical variables. This limits the comparability between both samples. The German sample (n = 1,664) was larger than the Ethiopian one (n = 256), however, this has an only marginal effect on the effect sizes for the comparison between Ethiopia and Germany. However, the differences in the distributions of the tumor stages can limit the comparability between the two samples. Because normative HADS data are not available for Ethiopia or Africa, it is difficult to estimate the level of burden due to the disease. The lack of strong measurement invariance for the comparison between Ethiopia and Germany limits the comparability of the mean scores obtained in these countries. Despite the problems with item D14, we do not recommend removing this item. A modified version omitting a problematic item would result in incomparability of the HADS results obtained in other examinations.

In conclusion, the Amharic version of the HADS provides reliable information on anxiety and depression. The high levels of mental burden in the Ethiopian cancer patients indicate unmet needs for psychosocial care.

## Supporting information

**S1 Data. HADS items.**
(SAV)

## Acknowledgments

We thank the patients, the study assistants who collected the data, and Mesfin Assefa who supervised the data collection.

## Author Contributions

**Conceptualization:** Yemataw Wondie, Andreas Hinz.

**Data curation:** Anja Mehnert.

**Formal analysis:** Andreas Hinz.

**Investigation:** Yemataw Wondie, Anja Mehnert.

**Writing – original draft:** Yemataw Wondie, Anja Mehnert, Andreas Hinz.

**Writing – review & editing:** Yemataw Wondie, Andreas Hinz.

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
