## [Decision Letter · Decision Letter 0]

21 Jul 2020

PONE-D-20-01559

The Hospital Anxiety and Depression Scale (HADS) applied to Ethiopian cancer patients

PLOS ONE

Dear Dr.Hinz ,

Thank you for submitting your manuscript to PLOS ONE. After careful consideration, we feel that it has merit but does not fully meet PLOS ONE’s publication criteria as it currently stands. Therefore, we invite you to submit a revised version of the manuscript that addresses the points raised during the review process.

Please address the issues raised regarding disparate sample sizes and how this problem was addressed in the analyses.

We look forward to receiving your revised manuscript.

Kind regards,

Rosemary Frey

Academic Editor

PLOS ONE

Journal Requirements:

2. Please upload a copy of Figure 2, to which you refer in your text on page 13. If the figure is no longer to be included as part of the submission please remove all reference to it within the text.

Reviewers' comments:

Reviewer's Responses to Questions

**Comments to the Author**

1. Is the manuscript technically sound, and do the data support the conclusions?

Reviewer #1: Partly

Reviewer #2: Yes

2. Has the statistical analysis been performed appropriately and rigorously? 

Reviewer #1: No

Reviewer #2: Yes

3. Have the authors made all data underlying the findings in their manuscript fully available?

Reviewer #1: No

Reviewer #2: Yes

4. Is the manuscript presented in an intelligible fashion and written in standard English?

Reviewer #1: Yes

Reviewer #2: Yes

5. Review Comments to the Author

Reviewer #1: The study is very interesting in describing the HADS-results in Ethiopian cancer patients.

Unfortunately, there are many aspects which should be considered.

A major revision is necessary.

(see attachment)

Reviewer #2: This is an important study that will make useful contributions to existing knowledge in related fields, especially with regard to Ethiopia. Given the cross-cultural comparisons, it might have been more appropriate to have submitted this to the Journal of Cross-Cultural Psychology for publication. However in all other respects it appears to be sound and useful. The manuscript is clearly written and logically coherent, the alphas of the related scales are sufficient, and the measures used are appropriate and valid, as well as the added check of comparison across cultural contexts. One can imagine that the disparate sizes of the Ethiopian and German samples made comparisons more difficult because the threshold for achieving significance would be different in the two samples, but this is not prohibitive. The final discussion section should acknowledge this limitation to the study however.

A few very minor specific recommended corrections...

35 - was/were. 'Data' is plural, so should be 'Data were...'.

50 - Add 'Germany' to keywords.

6. PLOS authors have the option to publish the peer review history of their article (what does this mean?). If published, this will include your full peer review and any attached files.

Reviewer #1: No

Reviewer #2: No

---

## [Decision Letter · Decision Letter 1]

5 Oct 2020

PONE-D-20-01559R1

The Hospital Anxiety and Depression Scale (HADS) applied to Ethiopian cancer patients

PLOS ONE

Dear Dr. Hinz,

Thank you for submitting your manuscript to PLOS ONE. After careful consideration, we feel that it has merit but does not fully meet PLOS ONE’s publication criteria as it currently stands. Therefore, we invite you to submit a revised version of the manuscript that addresses the points raised during the review process.

Please address the issues raised by Reviewer 1 concerning sample size differences.

We look forward to receiving your revised manuscript.

Kind regards,

Rosemary Frey

Academic Editor

PLOS ONE

Reviewers' comments:

Reviewer's Responses to Questions

**Comments to the Author**

1. If the authors have adequately addressed your comments raised in a previous round of review and you feel that this manuscript is now acceptable for publication, you may indicate that here to bypass the “Comments to the Author” section, enter your conflict of interest statement in the “Confidential to Editor” section, and submit your "Accept" recommendation.

Reviewer #1: (No Response)

Reviewer #2: All comments have been addressed

2. Is the manuscript technically sound, and do the data support the conclusions?

Reviewer #1: Yes

Reviewer #2: Yes

3. Has the statistical analysis been performed appropriately and rigorously? 

Reviewer #1: Yes

Reviewer #2: Yes

4. Have the authors made all data underlying the findings in their manuscript fully available?

Reviewer #1: Yes

Reviewer #2: Yes

5. Is the manuscript presented in an intelligible fashion and written in standard English?

Reviewer #1: Yes

Reviewer #2: Yes

6. Review Comments to the Author

Reviewer #1: Thank you for considering the comments.

Only two remaining aspects:

"... This limits the comparability between both samples. The German sample was larger than the Ethiopian one, however, this has an only marginal effect on the effect sizes for the comparison between Ethiopia and Germany. ..."

Please include the number of the german sample in the text and mention that the tumor stage was different in the Ethiopian versus the german sample (Stage I 7,8% versus 22,2%) which can influence the HADS results.

Reviewer #2: The sample size differences and other incomparable features between the two samples have now been acknowledged and discussed as study limitations in the concluding section, and in other respects the analysis seems adequate at this revised stage.

7. PLOS authors have the option to publish the peer review history of their article (what does this mean?). If published, this will include your full peer review and any attached files.

Reviewer #1: No

Reviewer #2: No

---

## [Author Response · Author response to Decision Letter 1]

7 Oct 2020

Reviewer #1: 

Please include the number of the german sample in the text and mention that the tumor stage was different in the Ethiopian versus the german sample (Stage I 7,8% versus 22,2%) which can influence the HADS results.

Response.

We already gave the number of the German sample in the methods section.

In the Limitations section we now repeated the number and acknowledged that the differences in the distributions of the tumor stages limit the comparability of the two samples.

Reviewer #2 had no additional remarks.

---

## [Decision Letter · Decision Letter 2]

20 Nov 2020

The Hospital Anxiety and Depression Scale (HADS) applied to Ethiopian cancer patients

PONE-D-20-01559R2

Dear Dr. Hinz,

We’re pleased to inform you that your manuscript has been judged scientifically suitable for publication and will be formally accepted for publication once it meets all outstanding technical requirements.

Kind regards,

Rosemary Frey

Academic Editor

PLOS ONE

Additional Editor Comments (optional):

Reviewers' comments:

Reviewer's Responses to Questions

**Comments to the Author**

1. If the authors have adequately addressed your comments raised in a previous round of review and you feel that this manuscript is now acceptable for publication, you may indicate that here to bypass the “Comments to the Author” section, enter your conflict of interest statement in the “Confidential to Editor” section, and submit your "Accept" recommendation.

Reviewer #1: All comments have been addressed

Reviewer #2: All comments have been addressed

2. Is the manuscript technically sound, and do the data support the conclusions?

Reviewer #1: (No Response)

Reviewer #2: Yes

3. Has the statistical analysis been performed appropriately and rigorously? 

Reviewer #1: (No Response)

Reviewer #2: Yes

4. Have the authors made all data underlying the findings in their manuscript fully available?

Reviewer #1: (No Response)

Reviewer #2: Yes

5. Is the manuscript presented in an intelligible fashion and written in standard English?

Reviewer #1: (No Response)

Reviewer #2: Yes

6. Review Comments to the Author

Reviewer #1: (No Response)

Reviewer #2: (No Response)

7. PLOS authors have the option to publish the peer review history of their article (what does this mean?). If published, this will include your full peer review and any attached files.

Reviewer #1: No

Reviewer #2: No

---

## [Editor Report · Acceptance letter]

24 Nov 2020

PONE-D-20-01559R2 

The Hospital Anxiety and Depression Scale (HADS) applied to Ethiopian cancer patients 

Dear Dr. Hinz:

I'm pleased to inform you that your manuscript has been deemed suitable for publication in PLOS ONE. Congratulations! Your manuscript is now with our production department. 

Kind regards, 

on behalf of

Dr. Rosemary Frey 

Academic Editor

PLOS ONE